# Infidelity in the Adolescence Stage: The Roles of Negative Affect, Hostility, and Psychological Well-Being

**DOI:** 10.3390/ijerph20054114

**Published:** 2023-02-25

**Authors:** Ana M. Beltrán-Morillas, Laura Villanueva-Moya, M. Dolores Sánchez-Hernández, María Alonso-Ferres, Marta Garrido-Macías, Francisca Expósito

**Affiliations:** 1Department of Social Psychology, University of Granada, 18071 Granada, Spain; 2Department of Social, Work and Differential Psychology, Complutense University of Madrid, 28040 Madrid, Spain; 3Department of Communication and Social Psychology, University of Alicante, 03690 Alicante, Spain

**Keywords:** adolescence stage, hostility, negative affect, unfaithful, well-being

## Abstract

Background: Infidelity is a relational process common in all types of romantic relationships and has been established as one of the main causes of relationship breakdown. However, little is known about this type of transgression in adolescent romantic relationships, although it manifests as a fairly frequent behavior involving different motivations. Even less is known about the emotional impact of infidelity on the offending person and its association with hostile behavior and psychological well-being. Methods: Through an experimental study (N = 301 Spanish adolescents (190 female and 111 male; *M*_age_ = 15.59, *SD* = 0.69; range from 15 to 17), we sought to analyze the effect of manipulating two types of motivations for infidelity (sexual vs. emotional dissatisfaction) on negative affect, hostility, and psychological well-being. Results: The main results revealed that committing infidelity motivated by hypothetical sexual (vs. emotional) dissatisfaction was indirectly related to lower psychological well-being through its effects on increased negative affect and hostility. Conclusions: Last but not least, we discuss these findings, highlighting the possible implications of infidelity for the psychosocial and psychosexual development of adolescents.

## 1. Introduction

Infidelity is a widespread phenomenon in the field of romantic relationships [1,2]. It is considered the most serious and threatening transgression in such relationships [3] as well as the main source of divorce and conjugal violence [2,4]. Nevertheless, although infidelity is well known to have a harmful effect on two members in a relationship and especially on those who suffer from it [3,4,5], little is known about the reasons for infidelity at the adolescent stage. Despite infidelity being fairly common behavior in this population [1,6,7], the main focus of study until now has been directed at the adult population [1,4].

As a way to approach the understanding of infidelity in adolescence, some research has shown emotional (e.g., distress, hurt, revenge, or indifference) [8] and behavioral indicators (e.g., physical, sexual, verbal, or emotional abuse) [9] that adolescents manifest or could externalize after suffering infidelity. However, the negative emotional impact from the perspective of the offending person and its relationship with hostile behavior and psychological well-being have not been addressed. Therefore, we aimed to shed light on the implications of infidelity in the adolescent stage and to analyze the relationship between different motivations for infidelity and negative affect, hostility, and psychological well-being.

### 1.1. Motivations for Infidelity in Adolescence: Sexual and Emotional Dissatisfaction

In empirical literature, infidelity is often referred to in terms such as adultery, deception, unfaithfulness, extramarital affairs, or dating someone other than one’s main partner [4,5], finding broad consensus on the part of the general population that sexual and emotional behaviors are the most constitutive of infidelity [5,10,11]. At this point, researchers have proposed multiple meanings to define this type of relational transgression. One of the definitions most used to refer to this concept points to infidelity as a violation of the implicit or explicit commitment previously agreed to by both members of the relationship, which may adopt a sexual, romantic, or mixed nature with a person besides the main partner [12]. However, it should be taken into consideration that the notion of infidelity may vary according to culture (e.g., while infidelity in Western countries such as Great Britain, the United States or Spain is not tolerated, Eastern countries such as Thailand have a more positive view of this behavior because sex has traditionally been commercialized) [13], or the type of relationship decreed by the members of the couple (e.g., polyamorous couples or couples that are not considered traditional and create agreements to carry out extradyadic behaviors) [14,15]. Similarly, biological sex can also influence acceptance of infidelity. Specifically, although it is generally men who tend to report greater involvement in extra-dyadic behaviors compared to women [16], this dissimilarity is diminishing in the younger population [17], which could be a reflection of recent societal changes showing more liberal attitudes toward sexuality [18]. Ultimately, the previous definition of infidelity would not be acceptable if one takes into account the diversity of opinions and existing judgments about the behaviors that are considered unfaithful, finding in the empirical literature disagreements from one person to another based on their participation or not in episodes of infidelity [11,19].

Historically, the incidence of infidelity in the world has been known to be over 60% [20], and such behavior is more likely between adolescents and young adults [6,21,22]. It is estimated that about 40–60% of adolescents are unfaithful to their partner, although around 70% of adolescents censure infidelity, particularly of a sexual kind [7]. Despite the high prevalence of infidelity in adolescence, however, study of this phenomenon during this stage is an aspect that has been largely neglected.

There are many theoretical meanings that have been proposed to approach what would make up the adolescent stage, finding among them a wide spectrum of discrepancies. For example, there is variability around the age at which adolescence begins, ranging from 12 to 25 years. In this regard, a thought that has been popularly spread among today’s society is to point out that today, young people begin adolescence very early and leave it very late [23]. Likewise, it has been suggested that adolescence can generally be conceived as a dynamic evolution that depends on physiological, psychosocial, temporal and cultural factors [24]. Furthermore, emphasis has been placed on its biological aspect, defining adolescence as a transition on a physical level marked by the onset of puberty and the end of physical growth; at the cognitive level, as a change in the ability to think abstractly; or at a social level, as a stage of preparation to acquire roles or adult maturity [25]. However, one of the definitions that has obtained the most consensus from experts in the area is the one proposed by the World Health Organization (WHO), which determines the adolescent stage as a phase of life that oscillates between 10 and the age of 19, and which would be characterized in parallel by physical growth, together with emotional, psychosocial and behavioral changes, originating in this way, the transformation from childhood to adulthood.

Adolescents thrive on the development of their identity and personal intimacy on the journey to adulthood [26]. Namely, during this period, their senses of identity, intimacy, and fidelity are closely connected [1,8], such that as their sense of personal identity is strengthened, their disposition toward intimacy and loyalty to their romantic partner increases [8,27]. That is, adolescents value the possibility of being faithful to their partners for the first time in their growth [7,8,21]. Nonetheless, sometimes these principles may not be in harmony with adolescents’ true intentions [1,28]. In other words, although the desire for intimacy may prompt adolescents to commit to a relationship, the desire to explore their personal identity may hinder them from bonding strongly with a relationship [1,8,21]. Consequently, committing infidelity during adolescence could be motivated as a result of trying to counter these intrinsic goals [1,8]. Thus, adolescents may be motivated to either create an emotional bond with a person other than their main partner or enjoy a greater plurality of sexual partners and/or sexual frequency without forming an emotional connection [1,8,21].

Empirical evidence has identified two types of motivations for infidelity that are mostly reported in adult romantic relationships. Such motivations refer to sexual and emotional dissatisfaction with the partner and the relationship [29,30,31,32]. Precisely, sexual dissatisfaction is encouraged by the desire to explore different sexual practices and to enjoy a greater plurality of sexual partners, whereas emotional dissatisfaction is caused by great disappointment and appreciation of a replacement partner’s positive qualities (e.g., finding a smarter alternative partner than the main partner) [29,30]. With special attention to the adolescent population, empirical literature has indicated that, like adults, adolescents seem to engage in infidelity for reasons of sexual dissatisfaction (e.g., desire to have sexual intercourse more frequently or sexual enjoyment) and emotional dissatisfaction with the partner and the relationship (e.g., neglect or emotional detachment) [1,8,21,33,34]. Therefore, sexual or emotional dissatisfaction may appear to be the key reason why adolescents can be unfaithful to their partners.

### 1.2. Being Unfaithful in Adolescence: Negative Affect, Hostility, and Psychological Well-Being

The empirical literature has focused mainly on examining the effects of infidelity on the offended person, neglecting to inquire into what infidelity entails for the offending person [35], and this gap is even wider in the adolescent stage [1,7]. This stage is quite lacking in this field of research because understanding the impact of infidelity on adolescent offenders is important for not only their relational but also their personal well-being. In this regard, we seek to analyze factors such as negative affect, hostility, and psychological well-being that may have a significant impact on adolescent offenders due to infidelity.

Negative affect has been referred to as the degree to which people peculiarly notice aversive and distressing emotions [36]. This negative affective state can appear in offenders after acknowledging their misconduct [20,37,38]. In this regard, if infidelity is noted, the perpetrator of the betrayal could feel strong emotional distress [37], especially if the infidelity is of a sexual nature, because such infidelity has been estimated as the most intransigent and causes greater distress in both the adult population [12,39] and the university-age and adolescent ones [3,7]. More specifically, if this last population were to be taken into account, the motivations for infidelity during adolescence (i.e., sexual and emotional dissatisfaction) could be weighed as internal and individual needs [1,40]. Such motivations could play a relevant role in balancing adolescents’ emotional levels, especially of negative affect. In this sense, although empirical works about the emotional impact of infidelity on teenage offenders are quite limited [1,8,21], research has shown that during this period, the individual need to experience different sexual practices or habits tends to increase, and the individual tends to have sexual intercourse more frequently [41]. That is, adolescents seem to feel a need mainly for sex beyond establishing an emotional bond with a partner [42]. Researchers have widely found that adolescent romantic relationships are characterized by being short-lived and presenting low levels of emotional intimacy in comparison to adult romantic relationships [43,44]. Therefore, if the previously reported findings are appreciated, emphasizing that adolescents appear to be stimulated primarily for sexual reasons, it would not be unusual for adolescents to experience high levels of negative affect when their reasons to engage in infidelity address sexual (vs. emotional) dissatisfaction. Thus, we expected to find that committing hypothetical infidelity due to sexual dissatisfaction would lead to higher levels of negative affect (Hypothesis 1a).

For its part, hostility has usually been specified as a set of beliefs and negative attitudes toward people, an expectation about others as the origin of inequalities, a relational perspective of opposing or showing rivalry to others, and a desire to harm others or see others hurt [45,46]. Although hostility has not been studied in isolation within the adolescent population, existing research has shown that adolescents exhibit aggressive behavior as a way of interacting with their peers [47,48]. This behavior, known as adolescence-limited antisocial behavior, tends to decrease in the course of adult life and is considered part of the normal developmental process during this period [47,49]. Within adolescent relationships, researchers have observed a gradually increasing interest toward sexual relations, which seems to include an emergent component in the way of relating to adolescent peers: showing rivalry for the conquest of a partner. That is, beyond integrating with their peers, adolescents also seem to fight for sexual dominance and status [44], a conflict found in both adolescent boys and girls [50,51,52]. In this sense, various empirical works have found that social behavior focused on these aspects seems to be determined by the emission of aggressive behaviors toward adolescents’ same-sex peers [44,53]. Although there has been no evidence of such an effect thus far, the aforementioned rivalry could intensify when adolescents experience sexual dissatisfaction in a relationship (e.g., when a partner does not want to innovate with different sexual practices or does not have sexual intercourse frequently) [29], which might dispose young people to perceive peers as a potential threat in the conquest or seduction of a partner [44,54]. This is why, despite the fact that adolescents could be motivated to commit infidelity due to sexual dissatisfaction with the partner, we expected that perceiving a threat to their dominance and sexual status [44] could be associated with increased levels of hostility (Hypothesis 1b).

Researchers have also demonstrated that negative affect is associated with greater inclinations and aggressive behaviors [55,56]. More specifically, from the frustration–aggression hypothesis [55,57] and self-determination theory [58], aggressive behavior tends to manifest when people experience frustration upon realizing that they are not successfully achieving their personal goals [57] or meeting their primary psychological needs for relational satisfaction, autonomy, and competence [58]. Furthermore, ref. [55], in a reformulation of the frustration–aggression theory, reasoned that frustrations “generate aggressive inclinations only to the extent that they produce negative affect” (p. 71). That is, negative affect is the foundation of aggressive inclinations, whereas frustration is only a potential reason for negative affect [59]. This tendency includes not only behavior but also more cognitive factors, such as hostility [55]. Therefore, by reasoning the above assumptions and considering sex as a primary need in adolescent development [42] as well as the low levels of emotional involvement that characterize romantic relationships established during that period [43,44], one would sensibly expect that committing infidelity motivated by sexual (vs. emotional) dissatisfaction would increase internal levels of hostility by prior association with high levels of negative affect.

Ultimately, psychological well-being has been determined as a positive consequence of personal perception on the proper development of life and of one’s potential [60], resulting in an optimal degree of psychological functioning [61]. Considering the adolescent population, it is noteworthy that optimal functioning is more than the absence of setbacks or pathologies [62]; thus, the study of well-being is a significant element during this stage of development [63]. Specifically, research has emphasized the significant roles of peer and romantic relationships for the development and psychological well-being of adolescents [62,64], given that those links are often perceived in terms of affiliation and passion, respectively [65]. Regarding the latter type of relationship, empirical literature has revealed that frequency or variety of sexual experiences [66,67]—even more so if they are performed with a romantic partner [68,69]—seem to promote higher levels of psychological well-being [70,71]. Thus, it would be reasonable to expect that engaging in an act of infidelity for reasons of sexual dissatisfaction with the main partner could raise low levels of psychological well-being (Hypothesis 1c).

Regarding the relationship between negative affect and psychological well-being, several studies have shown that negative affect is negatively associated with well-being in both adults [72] and youths and adolescents [73,74]. Especially in this last population, studies have found that the degree to which negative affect is associated with well-being seems to be linked to significant domains for adolescents (e.g., sexual skills and competencies) [75,76]. On a different note, studies have demonstrated that aggressiveness is associated with lower levels of well-being [77,78], although no empirical works have evidenced the relationship between hostility and psychological well-being, much less in the adolescent population. Instead, empirical evidence has supported that hostility negatively affects levels of physical well-being—an aspect that has been widely related to psychological well-being [79,80]—as a consequence of an individual’s negative interpersonal experiences [81,82]. Based on these arguments, we expected that high levels of negative affect (Hypothesis 2a) and high rates of hostility (Hypothesis 2b) would lead to lower levels of psychological well-being in the condition of infidelity due to sexual dissatisfaction.

However, according to what was mentioned in previous paragraphs, it is worth reasoning on the basis of the frustration–aggression hypothesis [55,57] and self-determination theory [58] because sexual activity with a partner is considered as a primary need in the adolescent period [42], and peer rivalry for dominance and sexual status also acquires meaning during this stage [44]. Hence, one can expect that infidelity motivated by sexual dissatisfaction could cause higher rates of negative affect, which would in turn be associated with high levels of hostility and ultimately with lower levels of psychological well-being (Hypothesis 3).

### 1.3. The Current Research

Although it is known that infidelity can cause detrimental effects on both members of a relationship [35], research on the consequences that this type of transgression can have on an offending person is deficient—and even more so in the adolescent period, despite the phenomenon occurring quite frequently [6,7,21]. Similarly, empirical studies that have addressed the issue of infidelity in the adolescent stage up to the present time have mainly involved the American population [7,8], despite infidelity having an equally negative connotation in the United States as in other Western countries such as Spain [5,13]. That is why it might be innovative to offer data from other countries such as Spain, where infidelity rates are estimated to be higher when compared with other countries in the European Union [5]. Because romantic relationships during adolescence seem to be a key element for the psychosocial development of those in that period [62,63], investigating indicators that this population could exhibit after committing infidelity is considered highly relevant. Thus, in the present study, we aimed to approximate the implications of infidelity in the adolescent stage, analyzing the association between different hypothetical motivations for infidelity and negative affect, hostility, and psychological well-being. It is noteworthy that numerous empirical works have focused on the field of infidelity using scenarios to simulate this relational transgression [3,39,83,84,85], considered a common methodology in the representation of social interactions [86].

In particular, we expected that committing a hypothetical infidelity due to sexual dissatisfaction would trigger higher levels of negative affect (Hypothesis 1a), higher levels of hostility (Hypothesis 1b), and lower levels of psychological well-being (Hypothesis 1c) than committing infidelity motivated by emotional dissatisfaction. Similarly, we also expected that high rates of negative affect (Hypothesis 2a) and high levels of hostility (Hypothesis 2b) would predict lower levels of psychological well-being. Finally, we examined whether committing infidelity motivated by sexual dissatisfaction (vs. emotional dissatisfaction) would elicit higher rates of negative affect, in turn associated with high levels of hostility and, consequently, with lower levels of psychological well-being (Hypothesis 3).

## 2. Materials and Methods

### 2.1. Participants

We recruited participants from four high schools belonging to the province of Granada (Spain). Using the G*Power 3.1 program [87], we estimated a sample size of 128 for a medium effect size of 0.5, significance level of 0.05, and power of 0.80, and we applied a statistical test to determine the difference between two independent means (two groups: sexual dissatisfaction vs. emotional dissatisfaction condition). Three hundred sixty-four people (138 male and 226 female) aged between 14 and 19 years (*M* = 15.64, *SD* = 0.77) took part in the study. We removed 63 of the participants from the analysis because they did not properly respond to the manipulation check of the experimental condition. Therefore, the final sample consisted of 301 adolescents (111 male and 190 female), who ranged in age from 15 to 17 (*M* = 15.59, *SD* = 0.69). All participants were Spanish. Of this sample, 25.2% reported maintaining a dating relationship at the time of the study, whereas 74.8% reported not having a relationship. Specifically, they were asked the following: are you currently in a dating relationship? (Yes/No). Moreover, about 17.9% professed that they had ever been unfaithful to their partner or ex-partner. They were asked the following: have you ever been unfaithful to your partner or ex-partner? (Yes/No). Participants who indicated “yes” also stated the type of infidelity: sexual nature or emotional/affectionate. A majority of the infidelity cases were of a sexual nature (55.6%), 33.3% were emotional or affectionate, and 11.1% had both natures.

### 2.2. Design and Procedure

Convenience sampling was employed to recruit potential participants from all schools located in Granada (Spain). First, we asked a public educational agency in the province to inform the high schools about the possibility of participating in the study. Specifically, this public agency sent a brief overview of the study to inform them of the main objectives of the research and to encourage them to collaborate. Secondly, the school counselors of the interested schools wrote to us to inform us of their willingness to participate in the study. Once their acceptance was obtained, school counselors sent to the parents a brief overview of the study’s intention and an informed consent form, through which they authorized their children to participate in the research. In the informed consent, parents were told about the anonymity and confidentiality of their children’s responses. All participants obtained informed parental consent in accordance with the Declaration of Helsinki. Three pretrained researchers administered the questionnaire to the participants in their classroom within a single session (i.e., 60 min). At the beginning of the session, the researchers read aloud the overall objective of the study and the instructions they had to follow to complete it. The researchers were present during the administration of the questionnaire to clarify possible doubts, ensure the anonymity of answers, and verify the independent completion of the participants. The questionnaire took approximately 20–30 min to complete. Once the questionnaire was filled out, the participants were informed about the true nature of the study, asked for feedback, and thanked for their participation. The research is part of an extensive project approved by the Ethics Committee of the University of Granada.

We used a factorial between-subjects design with an independent variable manipulated at two levels (sexual vs. emotional dissatisfaction). The participants were randomly presented a hypothetical situation of infidelity, which varied according to the reasons for committing it. Before introducing the experimental condition, the researchers showed participants a conceptual definition of infidelity to clarify possible doubts about it. The participants were then encouraged to envision about being unfaithful to their partner due to either sexual or emotional dissatisfaction and were asked to imagine their reactions to hypothetical situations [88]. Those who were not in a relationship had to try to imagine themselves in such a situation. The scenarios were developed based on motivations concerning sexual and emotional dissatisfaction taken from the Motivations for Infidelity Inventory (for further review, see [29]). After reading the infidelity situation, participants completed the measures of negative affect, hostility and psychological well-being.

### 2.3. Instruments

#### 2.3.1. Experimental Manipulation

Before introduction to the experimental manipulation, participants were shown a conceptual definition of infidelity to clarify possible doubts. This definition was based on one of the most commonly used in the infidelity field [12], as mentioned above. Then, to present the experimental manipulation, we developed two hypothetical situations to show various reasons for committing infidelity (sexual vs. emotional dissatisfaction) [20]. Specifically, in the condition of infidelity due to sexual dissatisfaction, adolescents were urged to envision that they were unfaithful because their partner did not want to have sex frequently, had lost interest in sex, and was unwilling to innovate sexual practices. Instead, in the condition of infidelity due to emotional dissatisfaction, adolescents had to imagine that they committed infidelity because their partner was emotionally distant, showed no interest in spending time together, and neglected their needs. Next, the participants responded to measures of negative affect, hostility, and psychological well-being.

#### 2.3.2. Negative Affect

The negative affect subscale of the Positive and Negative Affect Schedule [89,90] was used to assess the type of emotions that participants would experience if they were unfaithful to their partner (e.g., “upset”, “guilty”, “nervous”). This subscale is made up of 10 items and consists of a 5-point Likert-type response format ranging from 1 (*never*) to 5 (*very much*). The negative affect subscale has shown adequate psychometric properties in its adaptation to the Spanish adolescent population, with a reliability index ranging from 0.74 (young men) to 0.75 (young women). For this screened sample, we obtained a Cronbach’s alpha of 0.84.

#### 2.3.3. Hostility

The hostility subscale of Aggression Questionnaire [91,92] was used, which focuses on negative assessments that people have of others and of things, often accompanied by a clear desire to cause harm [45,46]. The hostility subscale contains eight items (e.g., “I sometimes feel like a powder keg ready to explode”) with a 5-point Likert scale (1 = *completely false to me*; 5 = *completely true to me*). This subscale has shown adequate psychometric properties in its validation with Spanish adolescents (Cronbach’s α = 0.65). In this sample, we obtained an alpha coefficient of 0.61.

#### 2.3.4. Psychological Well-Being Scale

The Psychological Well-Being Scale [93,94] consists of 29 items that assess how well people manage life challenges in six different domains, which refer to *self-acceptance* (four items; e.g., “In general, I feel confident and positive about myself”), *positive relations* (five items; e.g., “I know that I can trust my friends, and they know they can trust me”), *autonomy* (six items; e.g., “I have confidence in my opinions, even if they are contrary to the general consensus”), *environmental mastery* (five items; e.g., “The demands of everyday life often get me down”), *purpose in life* (five items; e.g., “I enjoy making plans for the future and working to make them a reality”), and *personal growth* (four items; e.g., “For me, life has been a continuous process of learning, changing, and growth”). Participants rated agreement with each of the items on a 6-point Likert scale (1 = *strongly disagree*; 6 = *strongly agree*). These subscales were organized into a second-order factor that denominates general psychological well-being [93], showing adequate psychometric properties in the Spanish population [95]. In the present sample, we obtained a reliability index of 0.85 for the total scale score.

#### 2.3.5. Manipulation Check

A manipulation check item was designed to test whether the participants had perceived the experimental condition as intended and therefore answered the previous measures in consideration of the scene they had just read (in relation to the situation you read at the beginning, “you were involved” in [a] sexual infidelity or [b] emotional or affectionate infidelity). Before statistical analyses, we removed all participants who answered the experimental manipulation incorrectly.

#### 2.3.6. Sociodemographic Characteristics

Data on sex, age, current relationship status, incidence of infidelity, and type of infidelity consummated were collected.

### 2.4. Statistical Analysis Strategy

We first conducted a chi-square test to check whether the experimental manipulation worked as intended. Next, to assess whether the motivation for committing infidelity (sexual vs. emotional dissatisfaction) elicited higher levels of negative affect and hostility and lower levels of psychological well-being, we performed a multivariate analysis of covariance (MANCOVA), including as covariates sex, age, participant relationship status, and incidence of infidelity. Subsequently, we implemented a multiple linear regression analysis to test whether in the condition of infidelity due to sexual dissatisfaction (vs. emotional dissatisfaction), high levels of negative affect and hostility would predict lower levels of psychological well-being (see Figure 1). Ultimately, we conducted a serial mediation analysis using PROCESS (Version 2; Model 6 [96]), to determine on the basis of negative-affect and hostility rates, the indirect effects of sexual dissatisfaction on psychological well-being (Table 1 and Figure 2). We introduced the condition of infidelity (sexual vs. emotional dissatisfaction) as the predictor variable (X), psychological well-being as the criterion (Y), and negative affect (M1) and hostility (M2) as the mediators. Sex, age, participant relationship status, and incidence of infidelity were included as covariates.

## 3. Results

### 3.1. Manipulation Check

The results confirmed the adequacy of the experimental manipulation in that 78.6% of the participants who read the condition of “sexual dissatisfaction” correctly identified it, whereas 86.3% of the participants who read the condition of “emotional dissatisfaction” recognized it as such, *χ^2^* (1, 364) = 155.07, *p* < 0.001, *φ* = 0.65. The remaining percentages, comprising participants in both conditions who responded poorly to the experimental manipulation, were eliminated.

### 3.2. Effects of Infidelity Motivations on Negative Affect, Hostility, and Psychological Well-Being

We conducted a MANCOVA to examine whether the act of infidelity due to sexual dissatisfaction (vs. emotional dissatisfaction) caused higher levels of negative affect (Hypothesis 1a), higher levels of hostility (Hypothesis 1b), and lower levels of psychological well-being (Hypothesis 1c). In these analyses, the infidelity condition was included as the independent variable, and negative affect, hostility, and psychological well-being were included as dependent variables. Sex, age, relationship status, and the previous incidence of infidelity were introduced as covariates.

The results revealed that motivations for infidelity (sexual dissatisfaction vs. emotional dissatisfaction) influenced negative affect, *Wilks’ λ* = 0.972, *F*(1, 295) = 5.02, *p* = 0.026, η_p_^2^ = 0.017. That is, sexual dissatisfaction seemed to elicit higher levels of negative affect than emotional dissatisfaction (*M*_sexual_ = 3.71, *SD* = 0.89; *M*_emotional_ = 3.48; *SD* = 0.79). Conversely, the motivations for infidelity did not significantly affect levels of either hostility (*Wilks’ λ* = 0.972, *F*(1, 295) = 1.43, *p* = 0.233, η_p_^2^ = 0.005) or psychological well-being (*Wilks’ λ* = 0.972, *F*(1, 295) = 2.14, *p* = 0.145, η_p_^2^ = 0.007). These results partially supported Hypothesis 1.

As far as covariates are concerned, sex had a significant effect on negative affect, *Wilks’ λ* = 0.930, *F*(1, 295) = 21.43, *p* < 0.001, η_p_^2^ = 0.067, such that young women compared to young men showed higher scores on negative affect (*M*_women_ = 3.74, *SD* = 0.80; *M*_men_ = 3.30; *SD* = 0.83). Likewise, maintaining a relationship, *Wilks’ λ* = 0.977, *F*(1, 295) = 5.10, *p* = 0.025, η_p_^2^ = 0.017, and the previous incidence of infidelity, *Wilks’ λ* = 0.977, *F*(1, 295) = 28.54, *p* < 0.001, η_p_^2^ = 0.088, had a significant effect on negative affect. Thus, those participants who were maintaining a relationship (*M*_yesrelationship_ = 3.66, *SD* = 0.78; *M*_norelationship_ = 3.55, *SD* = 0.85) and those who had not been unfaithful (*M*_noinfatihful_ = 3.68, *SD* = 0.75; *M*_yesinfatihful_ = 3.09, *SD* = 1.01) showed higher scores on negative affect. We performed an additional MANCOVA including the type of infidelity committed as a covariate, which did not significantly affect any of the dependent variables.

### 3.3. Effects of Motivations for Infidelity, Negative Affect, and Hostility on Psychological Well-Being

We conducted a multiple linear regression analysis to test whether in the condition of infidelity due to sexual dissatisfaction (vs. emotional dissatisfaction), high rates of negative affect (Hypothesis 2a) and high levels of hostility (Hypothesis 2b) would predict lower levels of psychological well-being. We included infidelity motivation (0 = sexual dissatisfaction; 1 = emotional dissatisfaction), negative affect, and hostility as predictive variables and psychological well-being as the criterion variable. In addition, we introduced sex (0 = young men; 1 = young women), age, whether the participants were currently in a relationship (0 = yes; 1 = no), and previous incidence of infidelity (0 = yes; 1 = no) as control variables. Prior to conducting the corresponding analysis, we standardized all the scores, contrasting the effects of the control variables in the first step, the predictive variables in the second step, and the interaction effects regarding the motivations for infidelity in the third step.

The results revealed that hostility was predictive of lower levels of psychological well-being (*b* = −0.470, *t* = −9.06, 95% CI [−0.573, −0.369], Δ*f*^2^ = 0.226). That is, people with high levels of hostility seemed to show lower levels of psychological well-being. Additionally, the results showed a significant interaction effect between motivations for infidelity × hostility (*b* = 0.499, 95% CI [0.022, 0.437], Δ*f*^2^ = 0.012). To ease interpretation of the interaction effect, we used simple slope tests to examine whether the difference between slopes was statistically significant (Dawson, 2014). The results revealed that there was a significant difference between the slopes; that is, the relation between motivations for infidelity and psychological well-being was greater for participants with higher hostility (*t* = 15.18, *p* < 0.001) than for those with lower hostility (*t* = 10.29, *p* < 0.001). Therefore, as can be observed in Figure 1, in the condition of sexual dissatisfaction, high levels of hostility were predictive of lower psychological well-being compared with low levels of hostility. Regarding the emotional dissatisfaction condition, hostility seemed not to be predictive of psychological well-being (Figure 1).

**Figure 1 ijerph-20-04114-f001:**
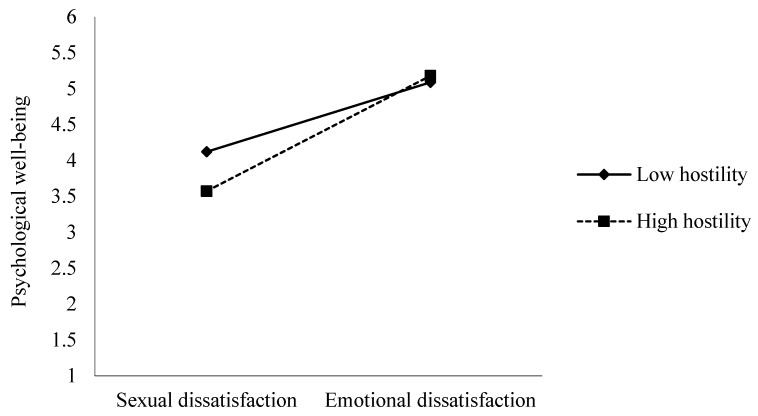
Interaction between motivations for infidelity and hostility on psychological well-being.

There were no simple effects of infidelity motivations (*b* = −0.054, *t* = −1.03, 95% CI [−0.206, 0.064], Δ*f*^2^ = 0.226) and negative affect on psychological well-being (*b* = −0.011, *t* = −0.198, 95% CI [−0.096, 0.078], Δ*f*^2^ = 0.226), or interaction effects between negative affect and infidelity motivations on psychological well-being (*b* = 0.002, *t* = 0.009, 95% CI [−0.160, 0.162], Δ*f*^2^ = 0.012).

These findings partially supported Hypothesis 2. The effects were obtained regardless of sex (*b* = −0.015, *t* = −2.64, 95% CI [−0.174, 0.133], Δ*f*^2^ = 0.012), age (*b* = −0.062, *t* = −1.05, 95% CI [−0.168, 0.051], Δ*f*^2^ = 0.012), whether the participants were currently in a relationship (*b* = −0.061, *t* = −1.02, 95% CI [−0.268, 0.085], Δ*f*^2^ = 0.012), and the previous incidence of infidelity (*b* = 0.080, *t* = 1.35, 95% CI [−0.063, 0.336], Δ*f*^2^ = 0.012).

### 3.4. Indirect Effect of Infidelity Motivations on Psychological Well-Being Based on Rates of Negative Affect and Hostility

Lastly, we examined whether committing infidelity motivated by sexual dissatisfaction (vs. emotional dissatisfaction) caused higher rates of negative affect, in turn associated with high levels of hostility and, consequently, with lower levels of psychological well-being (Hypothesis 3). We followed Hayes’s (2013) procedures to test indirect effects with serial mediators and considered bias-corrected confidence intervals for indirect associations based on 10,000 bootstrap samples. When a confidence interval (CI) does not include the value 0, it can be argued that there is a statistically meaningful association. Likewise, indirect effects can occur in the absence of a significant total effect [89]. Therefore, we used Model 6 of the PROCESS macro program (Version 2; Hayes, 2013) to observe the indirect effect of the infidelity motivations on psychological well-being based on rates of negative affect and hostility. Sex, age, relationship status, and incidence of infidelity were introduced as control variables (Table 1).

**Table 1 ijerph-20-04114-t001:** Multiple mediation analysis of the motivations for infidelity (sexual dissatisfaction vs. emotional dissatisfaction), negative affect, and hostility on psychological well-being.

Background	Negative Affect (NA)		Hostility (H)		Psychological Well-Being (PWB)	
	*Coeff.*	Symmetric BCI	*Coeff.*	Symmetric BCI	*Coeff.*	Symmetric BCI
Constant	4.919 ***	[2.609, 7.229]	1.706	[−0.030, 3.441]	6.344 ***	[4.613, 8.075]
Motivations for Infidelity (MI) ^a^	−0.202 *	[−0.385, −0.019]	0.115	[−0.036, 0.267]	−0.071	[−0.203, 0.062]
Negative Affect			0.117 *	[0.021, 0.214]	−0.009	[−0.103, 0.085]
Hostility					−0.471 ***	[−0.585, –0.357]
Sex ^b^	0.423 ***	[0.237, 0.609]	0.062	[−0.097, 0.221]	0.033	[−0.108, 0.175]
Age	−0.117	[−0.262, 0.027]	0.041	[−0.067, 0.149]	−0.038	[−0.147, 0.071]
Relationship Status ^c^	−0.238 *	[−0.461, −0.015]	−0.011	[−.208, 0.186]	−0.103	[−0.258, 0.052]
Infidelity Incidence ^d^	0.633 ***	[0.329, 0.937]	−0.164	[−0.395, 0.068]	0.102	[−0.101, 0.305]
	*R^2^* = 0.179		*R^2^* = 0.036		*R^2^* = 0.238	
	*F*(5, 295) = 11.58, *p* < 0.001		*F*(6, 294) = 2.05, *p* = 0.059		*F*(7, 293) = 12.13, *p* < 0.001	
Indirect Effects	Effects			Symmetric BCI		
Total	−0.041			[−0.123, 0.033]		
I1	0.002			[−0.015, 0.028]		
I2	0.011			[0.001, 0.033]		
I3	−0.054			[−0.131, 0.015]		

*Note*. I1 = MI→ NA →PWB; I2 = MI→ NA→ H →PWB; I3 = MI→ H→ PWB; Symmetric BCI: symmetric bootstrapping confidence interval; ^a^ sexual dissatisfaction = 0, emotional dissatisfaction = 1; ^b^ 0 = male, 1 = female; ^c^ 0 = yes, 1 = no; ^d^ 0 = yes, 1 = no. * *p* < 0.05, *** *p* < 0.001.

The variables included in the model predicted 23.8% of the variance of inclination to show psychological well-being. As can be seen in Table 1, sexual dissatisfaction (vs. emotional dissatisfaction) was indirectly linked to lower psychological well-being via its effects on increased negative affect and increased hostility. In particular, motivation to be unfaithful due to increased sexual dissatisfaction was associated with higher levels of negative affect, which in turn seemed to be related to higher levels of hostility and consequently to be associated with lower levels of psychological well-being (controlling for negative affect and hostility; Figure 2). The total effect of sexual dissatisfaction on psychological well-being was not significant (*b* = −0.112, *SE* = 0.08, 95% CI [−0.264, 0.039]). These findings supported Hypothesis 3. The covariates of sex, relationship status, and previous incidence of infidelity had a significant effect on negative affect. As indicated above, after we performed a MANCOVA, young women (vs. young men), people who were in a relationship (vs. people who had no relationship), and those who had not been unfaithful (vs. people who had been unfaithful) reported higher levels of negative affect.

We conducted complementary, different serial mediation analyses with each of the well-being dimensions, observing the indirect effect on all of them: self-acceptance (*b* = 0.015, *SE* = 0.01, 95% CI [0.002, 0.045]); positive relations (*b* = 0.013, *SE* = 0.01, 95% CI [0.002, 0.037]); autonomy (*b* = 0.012, *SE* = 0.01, 95% CI [0.001, 0.036]); environmental mastery (*b* = 0.011, *SE* = 0.01, 95% CI [0.001, 0.035]); purpose in life (*b* = 0.011, *SE* = 0.01, 95% CI [0.001, 0.032]); and personal growth (*b* = 0.005, *SE* = 0.00, 95% CI [0.001, 0.018]).

**Figure 2 ijerph-20-04114-f002:**
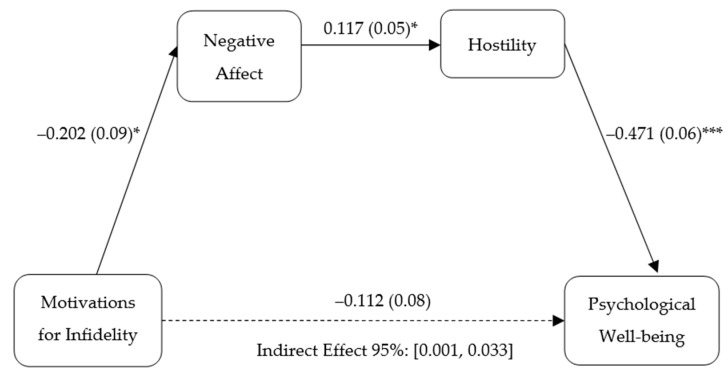
Conceptual model showing the indirect effect of multiple steps between motivations for infidelity and psychological well-being via negative affect and hostility. Unstandardized beta coefficients reported with standard errors within parentheses. Motivations for infidelity: 0 = sexual dissatisfaction, 1 = emotional dissatisfaction. * *p* < 0.05, *** *p* < 0.001.

## 4. Discussion

The purpose of this research was to examine how motivation to become involved in infidelity in adolescence (sexual vs. emotional dissatisfaction)—assessed through hypothetical relationship situations—could affect various emotional, behavioral, and well-being variables, such as negative affect, hostility, and psychological well-being.

First, the results showed that committing infidelity motivated by sexual dissatisfaction seems to elicit higher levels of negative affect compared to committing infidelity out of emotional dissatisfaction. As in the adult population, a high percentage of adolescents did not approve of infidelity of a sexual nature [7], because it comprises more explicit and less ambiguous behaviors than emotional infidelity; that is, sexual infidelity tends to be perceived as more constitutive of infidelity [5,19]. This appreciation could have elicited higher levels of negative affect, as participants might have perceived that engaging in such infidelity would be acting against their own moral code [7]. However, more research is required to infer such claims. Another explanation for this finding could be derived from the experimental manipulation, which encouraged adolescents to imagine themselves committing infidelity for reasons of either sexual or emotional dissatisfaction and subsequently evaluated their reactions to the prompt [88]. The adolescent period seems to be flooded by reasoning and response of a sexual nature, which highlights a primary need to have sexual intercourse that prevails over the establishment of an emotional connection [42]. This need could meet an intrinsic motivation focused on achieving a series of either physical (e.g., quest for pleasure or sexual satisfaction) or social goals to maintain status among peers and reinforce a sense of self (e.g., sexual competition or feeling of attractiveness) [75,97]. Thus, urging an adolescent to imagine that their partner has lost interest in sex, does not want to have sex as often, or does not show an incentive to innovate in sexual practices [29,66,67] might lead them to experience high levels of negative affect and undermine their levels of sexual self-efficacy and self-esteem when their sexual status and dominance are threatened. Future research could weigh this reflection and delve deeper into the connotations of infidelity in the adolescent period.

The results also evidenced that high levels of hostility were predictive of lower psychological well-being when infidelity was motivated by sexual dissatisfaction (vs. emotional dissatisfaction). The adolescent stage is usually marked by dominance and sexual status in a fight for the conquest of a partner, and researchers have observed a rivalry seeming to influence the way adolescents relate to their peers [44,53]. This rivalry seems to tend to increase when adolescents notice that their romantic partner is refusing or denying them aspects that they consider significant for their personal development [98], which appears to trigger levels of hostility and aggressiveness toward peers [99,100] even if they are not guilty of impeding the attainment of an adolescent’s personal goals [101]. Thus, adolescents’ tendency to perceive their peers as latent competition for persuading a partner [44,54] and bearing in mind that our study participants were asked to imagine getting involved in infidelity for sexual reasons (e.g., their partners had lost interest in having sex with them) [29] may have contributed to the promotion of high levels of hostility, ultimately lowering adolescents’ levels of psychological well-being as a result of this appreciation of their interpersonal experience [81,82]. Nevertheless, more research is needed to address the plausibility of this contribution.

Finally, the results revealed that carrying out infidelity due to sexual dissatisfaction (vs. emotional dissatisfaction) was associated with higher levels of negative affect, which in turn were associated with greater hostility, consequently resulting in lower levels of psychological well-being. The need to have sexual intercourse in the adolescent period seems to prevail over the establishment of an emotional bond [42]. It seems that the emotional connection with a partner begins to develop after having sexual intercourse with him or her [75,98]. Starting from the deference of the frustration–aggression hypothesis [55,57] and on the other from the theory of self-determination [57,58] this intrinsic imposition in the search for sexual pleasure [75,98] could lead adolescents to suffer high levels of negative affect when they experience frustration on account of having committed infidelity because their partner lost the incentive for sex with them [29,66,67]. At the same time, adolescents might also feel unsuccessful in their desire to maintain their social and sexual status vis à vis their peers (i.e., not meeting their psychological needs for relational satisfaction, competence, and autonomy [58]). Hence, appearing to be unfaithful to a partner for reasons of sexual dissatisfaction could trigger high negative affectivity, leading adolescents to manifest higher levels of hostility toward their peers when they perceive them as a potential threat in the recruitment or seduction of their partner, and consequently affecting their levels of psychological well-being. However, more research is required to pin down these issues and to examine the scapegoat for hostility (i.e., the hostile individual’s partner or peers) to better understand adolescent psychosocial and psychosexual development. It is also plausible that the impact of negative affect and hostility could trigger long-term adverse outcomes and influence the way adolescents behave with their sexual partners, as has been demonstrated in the adult population (e.g., in men, increased discomfort and hostility after an indirect request for condom use) [102]; adolescents could even contemplate the use of physical force or other forms of aggression (e.g., verbal aggression) to resolve the frustration and lack of well-being that both types of dissatisfaction—with special attention to the sexual kind—could cause for them. Nonetheless, caution should be exercised, as more research is needed to address these complex issues.

We consider it noteworthy that young women, adolescents who were in a relationship, and those who had not previously been unfaithful showed higher scores on negative affect. Usually, women have tended to exhibit a greater willingness to care for and maintain a relationship [103,104]; thus, a conflict or betrayal that violates those principles—such as engaging in infidelity—could arouse a deep negative affect in them, although such a relationship would probably be over [103]. Adolescence is a period of searching for one’s identity and of personal exploration [8], and romantic relationships established during adolescence are considered substantial for positive development during that stage [105]. In this sense, empirical evidence has shown that sexual intimacy with a partner seems to promote higher rates of welfare and sexual satisfaction compared to intimacy with casual partners [68,69]. Thus, adolescents who have a relationship could present higher scores of negative affect when imagining being unfaithful to their partner. Finally, having no prior involvement in previous infidelity could provoke higher levels of negative affect because, as [7] pointed out, such people could perceive that they are contravening their ethical code if they commit infidelity. Therefore, although people may feel somewhat dissatisfied with their relationship, that feeling does not indicate that they are unfaithful to their partner [29]. Future studies could consider these issues to enlighten the arguments above.

Although this study offers data in the desired direction and we encourage further work in this field of research, there were some limitations that should be addressed in future research. Because the methodology used was based on hypothetical situations that devised infidelity, it would seem reasonable to question the extent to which the scenarios achieved the naturalness, experience, and objectivity of a real situation. Although it is considered inappropriate, different research domains have used this methodology to devise social interactions [86]. We should likewise note that a high rate of participating adolescents did not have a partner, which could have made it difficult for them to imagine themselves as participants in a situation of infidelity. For this reason, future studies should control for (with a direct question, for example) if the participants are really capable of imagining themselves in that situation with a hypothetical partner. In the same sense, not having a partner could have caused uncertainty in thinking about emotional or sexual dissatisfaction as reasons to engage in infidelity. However, because reasons to become involved in infidelity seem to be an intrinsic process [29], we do not consider this rate to be a major drawback of our study. Overall, our findings are a prelude for future research to investigate more deeply whether sexual or emotional dissatisfaction as motivations to commit infidelity during adolescence are affected by relationship status. It would be interesting for future studies to consider replicating this study in other Western countries or populations of different cultures to see how generalized our findings might be. Future empirical works might also delve more deeply into the nature of adolescent romantic relationships to determine if the associations found in our study could be extrapolated beyond heterosexual relationships and if they would differ, in turn, within LGTBIQ relationships. Ultimately, future studies could assess the roles of possible moderating variables such as the degree of emotional involvement and previous sexual experiences with a partner—although research has shown that such variables seem to promote high levels of well-being (e.g., high levels of self-esteem, lower isolation rates, or better self-image) [70,71]—and of personality traits such as narcissism, as narcissists seem to manifest higher levels of aggression and hostility when they are warned of threats to their ego [106].

## 5. Conclusions

This research approaches the understanding of infidelity in the adolescent stage and provides suggestive data for this field of research. The main results show that sexual dissatisfaction (vs. emotional dissatisfaction) seems to trigger high levels of negative affect, which in turn appear to be associated with high levels of hostility and ultimately result in worse psychological well-being. During the adolescent period, sexual practices or relationships seem to be a key element for adolescents’ psychosexual development. Despite being a preamble to future research, our findings could encourage clinical psychologists working with adolescents to focus on improving their emotional management and equipping them with the necessary skills to cope constructively with sexual dissatisfaction in a couple’s relationship. Likewise, from our work, we derive the need to develop sexual psychoeducation programs aimed at promoting tools that allow for buffering of the negative affect derived from sexual dissatisfaction and its possible consequences, for instance, working on aspects such as exploration of sexual attitudes, assertive communication, emotional intelligence, resilience in the face of sexual pressure created by social expectations, and conflict resolution strategies within relationships. We consider it essential to combine practical efforts in this direction because sexual behavior during adolescence is essential to achieving adequate social functioning and identity development and to acquiring greater competition in subsequent romantic relationships.

## Data Availability

The data that support the findings of this study are available from the corresponding author upon reasonable request.

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
