# Peer review of "Infidelity in the Adolescence Stage: The Roles of Negative Affect, Hostility, and Psychological Well-Being"

_ijerph, 2023, doi:10.3390/ijerph20054114_

Round 1

Reviewer 1 Report

This is an interesting study investigating infidelity among adolescents. Although the sample size was not big, this research addressed an interesting topic which has some theoretical and practical contributions. I have several comments:

1.      In line 31, “Nevertheless, although”. “although” should be “though”. Please check your grammar errors.

2.      I recommend the authors to propose the hypotheses separately in the literature review rather than in the section 1.3, so that the readers could easily follow your argument.

3.      Most of the participants (74.8%) were not in a relationship and had to “try to imagine themselves in such a situation”. This could be a great limitation of this study since we cannot control how the participants imagine. This should be carefully discussed in discussion chapter.

4.      There is an incomplete sentence in 2.3.1, “This definition was based on [12], one of the most commonly used in the infidelity field, as mentioned above”.  Please specify Based on what ?

5.      Please further explain the negative path from motivations for Infidelity to negative affect in Figure 2. It would be better for the readers if you clearly explain how “Motivations for Infidelity” was scored in the text not just in the figure notes

6.      I think the content in the footnote on page 10 could be involved in the text.

Author Response

Thank you so much for your comments, they were extremely helpful in improving our paper. We have revised the paper to address all of the suggestions you made, as detailed below. We greatly appreciate your time and thought in making this manuscript stronger. 

Below we include the comments you provided in entirety; the comments are italicized, followed by our responses, which are not italicized. The revisions to the manuscript are marked up using the "track changes". 

-------------------------------------------------------------------------------------

This is an interesting study investigating infidelity among adolescents. Although the sample size was not big, this research addressed an interesting topic which has some theoretical and practical contributions. I have several comments:

1. In line 31, “Nevertheless, although”. “although” should be “though”. Please check your grammar errors.

RESPONSE: Thank you so much for catching the mistake. We have corrected it in the manuscript

2. I recommend the authors to propose the hypotheses separately in the literature review rather than in the section 1.3, so that the readers could easily follow your argument.

RESPONSE: Following your recommendation, we have included the hypotheses separately in the literature review to facilitate the reading of our argument. However, we also decided to keep the hypotheses at the end of the introduction, with the aim of making it easier to remember these hypotheses after reading the entire literature review. 

3. Most of the participants (74.8%) were not in a relationship and had to “try to imagine themselves in such a situation”. This could be a great limitation of this study since we cannot control how the participants imagine. This should be carefully discussed in discussion chapter.

RESPONSE: Thank you for your suggestion. Although we mentioned this limitation in the discussion chapter (please see lines 602-608, p.15), we have included a broader discussion of this limitation and possible improvements in future studies in the discussion section.

4. There is an incomplete sentence in 2.3.1, “This definition was based on [12], one of the most commonly used in the infidelity field, as mentioned above”.  Please specify Based on what?

RESPONSE: Sorry for the misunderstanding. We referred to the definition of infidelity from Dillow et al. (2011). We have modified the order of the citation in the text to be clearer.

5. Please further explain the negative path from motivations for Infidelity to negative affect in Figure 2. It would be better for the readers if you clearly explain how “Motivations for Infidelity” was scored in the text not just in the figure notes

RESPONSE: The negative path from motivations for infidelity to negative affect is already described in the manuscript (please see lines 485-486, p.13). In particular, motivation to be unfaithful due to increase sexual dissatisfaction was associated with higher levels of negative affect.

6. I think the content in the footnote on page 10 could be involved in the text.

RESPONSE: Following your suggestion, we have included the content of such footnote in the text. Specifically, we have included that information when we finish explaining the indirect effect of Infidelity Motivations on Psychological Well-Being (please see lines 493-498, p.13).

Reviewer 2 Report

The authors begin their paper by pointing out all the critical issues that are related to infidelity. However, they adopt an essentially Eurocentric and traditionalist vision.

In many cultures infidelity is considered an acceptable and accepted social phenomenon. Furthermore, depending on the context, female and male infidelity can have very different social acceptability.

Finally, recent studies are focusing on the new forms of having a family, also considering polyamorous relationships or extended families, in which infidelity is taking on the characteristics of an ethically shared practice.

Consequently, I believe that the authors should broaden the literature review, developing a much more critical, complex and topical discourse around infidelity.

An analogous discourse can be made in relation to adolescence. When can subjects be defined as adolescents? Are there any generational differences? Are there different ways of living and understanding adolescence in different cultures? The sample includes people aged 13 to 19. Can they all really be considered adolescents?

I believe that these are crucial aspects to be better developed in a scientific article.

Regarding the method, the authors should clarify several issues.

For example, how did they get in touch with schools? Why did they choose to involve some schools and not others in the study?

Furthermore, they only indicated the number of participants. This information is partial. Readers should know at least the main socio-demographic characteristics of the sample. How are the participants distributed among the age groups? Between genres? By ethnicity?

Finally, the promise to underline the importance of the possible implications of infidelity for the psychosocial and psychosexual development of adolescents is only partially kept. I believe that the conclusions are quite inconsistent and should be further strengthened.

Author Response

Thank you so much for your comments, they were extremely helpful in improving our paper. We have revised the paper to address all of the suggestions you made, as detailed below. We greatly appreciate your time and thought in making this manuscript stronger. 

Below we include the comments you provided in entirety; the comments are italicized, followed by our responses, which are not italicized. The revisions to the manuscript are marked up using the "track changes". 

1. The authors begin their paper by pointing out all the critical issues that are related to infidelity. However, they adopt an essentially Eurocentric and traditionalist vision. In many cultures infidelity is considered an acceptable and accepted social phenomenon. Furthermore, depending on the context, female and male infidelity can have very different social acceptability. Finally, recent studies are focusing on the new forms of having a family, also considering polyamorous relationships or extended families, in which infidelity is taking on the characteristics of an ethically shared practice. Consequently, I believe that the authors should broaden the literature review, developing a much more critical, complex and topical discourse around infidelity.

RESPONSE: Thank you very much for the precision of your comment. We have considered it, and based on it, we have specified in each of the aspects that you indicate. Additional information follows the definition of infidelity.

2. An analogous discourse can be made in relation to adolescence. When can subjects be defined as adolescents? Are there any generational differences? Are there different ways of living and understanding adolescence in different cultures? The sample includes people aged 13 to 19. Can they all really be considered adolescents?

I believe that these are crucial aspects to be better developed in a scientific article.

RESPONSE: Thank you for transmitting your concern regarding the notion of adolescence. We have valued your comment and included additional information about adolescence. Said information is found in the third paragraph of the section “1.1. Motivations for Infidelity in Adolescence: Sexual and Emotional Dissatisfaction.

3. Regarding the method, the authors should clarify several issues.

For example, how did they get in touch with schools? Why did they choose to involve some schools and not others in the study?

RESPONSE: We are grateful for these remarks, we have indicated it in the design and procedure section. Specifically, it now said (please see lines 282-293, p.6): “Convenience sampling was employed to recruit potential participants from all schools located in [blinded for review]. First, we asked a public educational agency in the province to inform the high schools about the possibility of participating in the study. Specifically, this public agency sent a brief overview of the study to inform them of the main objectives of the research and to encourage them to collaborate. Secondly, the school counselors of the interested schools wrote to us to inform us of their willingness to participate in the study. Once their acceptance was obtained, school counselors sent to the parents a brief overview of the study’s intention and an informed consent form, through which they authorized their children to participate in the research.”

4. Furthermore, they only indicated the number of participants. This information is partial. Readers should know at least the main socio-demographic characteristics of the sample. How are the participants distributed among the age groups? Between genres? By ethnicity?

RESPONSE: As we indicated in lines 274-275 (p.6): “(…) the final sample consisted of 301 adolescents (111 male and 190 female), who ranged in age from 15 to 17 (M = 15.59, SD = 0.69)”. We did not ask students about their ethnicity, but we asked them about their nationality. We have added information about nationality in this description.

5. Finally, the promise to underline the importance of the possible implications of infidelity for the psychosocial and psychosexual development of adolescents is only partially kept. I believe that the conclusions are quite inconsistent and should be further strengthened.

RESPONSE: Thank you for your comment. We have worked on the conclusions of our study to provide greater consistency and clarity (see Conclusion section; page 16).

Round 2

Reviewer 2 Report

I really appreciated that authors addressed all of the suggestions I made.

In the current form, I think the paper is suitable for publication.